# NPTC-net: Narrow-Band Parallel Transport Convolutional Neural Network on Point Clouds

## Abstract

Convolution plays a crucial role in various applications in signal and image processing, analysis and recognition. It is also the main building block of convolution neural networks (CNNs). Designing appropriate convolution neural networks on manifold-structured point clouds can inherit and empower recent advances of CNNs to analyzing and processing point cloud data. However, one of the major challenges is to define a proper way to "sweep" filters through the point cloud as a natural generalization of the planar convolution and to reflect the point cloud's geometry at the same time. In this paper, we consider generalizing convolution by adapting parallel transport on the point cloud. Inspired by a triangulated surface based method Schonsheck et al. (2018), we propose the Narrow-Band Parallel Transport Convolution (NPTC) using a specifically defined connection on a voxelized narrow-band approximation of point cloud data. With that, we further propose a deep convolutional neural network based on NPTC (called NPTC-net) for point cloud classification and segmentation. Comprehensive experiments show that the proposed NPTC-net achieves similar or better results than current state-of-the-art methods on point clouds classification and segmentation.

## 1 Introduction

Convolution is one of the most widely used operators in applied mathematics, computer science and engineering. It is also the most important building block of Convolutional Neural Netowrks (CNNs) which are the main driven force in the recent success of deep learning LeCun et al. (2015); Goodfellow et al. (2016).

In the Euclidean space $\mathbb{R}^n$, the convolution of function $f$ with a kernel (or filter) $k$ is defined as

$$(f * k)(x) := \int_{\mathbb{R}^n} k(x - y) f(y) dy. \tag{1}$$

This operation can be easily calculated in Euclidean spaces due to the shift-invariance of the space so that the translates of the filter $k$, i.e. $k(x-y)$ is naturally defined. With the rapid development of data science, more and more non-Euclidean data emerged in various fields including network data from social science, 3D shapes from medical images and computer graphics, data from recommending systems, etc. Therefore, geometric deep learning, i.e. deep learning of non-Euclidean data, is now a rapidly growing branch of deep learning Bronstein et al. (2017). In this paper, we will discuss how we can generalize the definition of convolution to (manifold-structured) point clouds in a way that it inherits desirable properties of the planar convolution, thus it enables to design convolutional neural networks on point clouds.

One of the main challenges of defining convolution on manifolds and point clouds (a discrete form of manifolds) is to define translation $x - y$ on the non-Euclidean domain. Other than convolutions, we also need to properly define pooling to enable networks to extract global features and to save memory during training. Multiple types of generalization of convolutions on manifolds, graphs and point clouds have been proposed in recent years. We shall recall some of them and discuss the relation between existing definitions of convolutions and the proposed narrow-band parallel transport convolution.

## 1.1 RELATED WORK OF CONVOLUTIONS ON NON-EUCLIDEAN DOMAINS

**Spectral methods** avoid the direct definition of translation $x - y$ by utilizing the convolution theorem: for any two functions $f$ and $g$, $\widehat{f * g} = \hat{f} \cdot \hat{g}$. Therefore, we have $f * g = (\hat{f} \cdot \hat{g})^\vee$, where $\wedge$ and $\vee$ represent generalized Fourier transform and inverse Fourier transform provided through the associated Laplace-Beltrami (LB) eigensystem on manifolds. To avoid computing convolution through full eigenvalue decomposition, polynomial approximation is proposed and yields convolution as action of polynomials of the LB operator Hammond et al. (2011); Dong (2017). Thus, convolutional neural networks can be designed Bruna et al. (2014); Defferrard et al. (2016); Levie et al. (2019). Spectral methods, however, suffer two major drawbacks. First, these methods define convolution in the frequency domain. As a result, the learned filters are not spatially localized. Secondly, spectral convolutions are domain-dependent as deformation of the ground manifold will change the corresponding LB eigensystem. This obstructs the use of learning networks from one training domain to a new testing domain Schonsheck et al. (2018).

**Spatial mesh-based methods** are more similar to the Euclidean case, and this is one of the reasons why most of the existing works fall into this category. The philosophy behind these methods is that the tangent plane $\mathcal{T}_x\mathcal{M}$ of a 2-dimensional manifold $\mathcal{M}$ in each point $x \in \mathcal{M}$ is embedded to a 2-dimensional Euclidean domain where convolution can be easily defined. In this paper, we make the first attempt to interpret some of the existing mesh-based methods in a unified framework. We claim that most of the spatial mesh-based methods can be formulated as

$$(f * k)(x) := \int_{\mathcal{T}_{x,\epsilon}\mathcal{M}} k(\phi(x,v))f(v)\mathrm{d}v, \quad x \in \mathcal{M}, v \in \mathcal{T}_x\mathcal{M}. \tag{2}$$

Here, $k : \mathbb{R}^2 \to \mathbb{R}$ is a convolution kernel and $\mathcal{T}_{x,\epsilon}\mathcal{M} = \{v \in \mathcal{T}_x\mathcal{M} : \langle v, v \rangle_{g_x} \leq \epsilon^2\}$ with $\epsilon > 0$ being the size of the kernel. The mapping $\phi(x,v) : \mathcal{M} \times \mathcal{T}_x\mathcal{M} \to \mathbb{R}^2$ is defined as

$$\phi(x,v) = (\alpha_1, \alpha_2) = \left( \langle v, \vec{u}_x^1 \rangle_{g_x}, \langle v, \vec{u}_x^2 \rangle_{g_x} \right), \tag{3}$$

where $\vec{u}_x^j \in \mathcal{T}_x\mathcal{M}$, $j = 1, 2$. For simplicity, we will denote $\vec{u}_x = (\vec{u}_x^1, \vec{u}_x^2)$. Most of the designs of the existing manifold convolutions focused on the designs of $\vec{u}_x$. We remark that possible singularities will lead to no convolution operation at those points. These are isolated points on a closed manifold and for more discussion of singularities, please refer to the Appendix.

For example, GCNN Masci et al. (2015) and ACNN Boscaini et al. (2016) construct a local geodesic polar coordinate system on a manifold, formulating the convolution as

$$(f * k)(x) = \int k\left( (\phi \circ \psi)(\theta, r) \right) \left( \mathcal{Q}_x f \right)(r, \theta)\mathrm{d}r\mathrm{d}\theta,$$

where $\mathcal{Q}$ is a local interpolation function with interpolation domain an isotropic disc for GCNN and an anisotropic ellipse for ACNN. A local geodesic polar coordinate system on a manifold can also be transformed to a 2-dimensional planar coordinate system on its tangent plane. Such transformation is the mapping $\psi$ which is defined by the inverse exponential map: $v = exp^{-1}(z(\theta, r))$ with $z(\theta, r)$ being a point in the local geodesic polar coordinate system at $x \in \mathcal{M}$ with coordinates $(\theta, r)$. With this, we can easily interpret ACNN within the framework of (2). Indeed, ACNN essentially chooses $\vec{u}_x$ as the directions of the principal curvature at point $x$. For GCNN, on the other hand, it avoids choosing a specific vector field on the manifold by taking max-pooling among all possible directions of $\vec{u}_x^1$ at each point. Such definition of convolution, however, ignores the correspondence of the convolution kernels at different locations.

The newly proposed PTC Schonsheck et al. (2018) defines convolution directly on the manifold, while using tangent planes to transport kernels by a properly chosen parallel transport. PTC can be equivalently cast into the form of (2) using the inverse exponential map, and implementation of the proposed parallel transported is realized through choosing specific vector fields $\{\vec{u}_x^j\}_{x \in \mathcal{M}}, j = 1, 2$ guided by a Eikonal equation for transforming vectors along geodesic curves on manifolds.

**Spatial point-based methods** have wider applications due to their weaker assumptions on the data structure, a point cloud $\mathcal{P} = \{x_i \in \mathbb{R}^3 : i = 1, \ldots, N\} \subset \mathbb{R}^3$ consists of points in a 3-dimensional Euclidean space with the coordinates of the points as the only available information. Manifolds can be approximated by point clouds via sampling. Computing $k$-nearest neighborhoods of $x$ or

neighbors within a fixed radius can easily convert a point cloud to a local graph or mesh. This is why point cloud is simple, flexible and attracts much attentions lately.

There are mainly two types of point-based convolution. The first type is to combine the information of points directly. These methods can be formulated as

$$(f * k)(x_i) := \sum_{x_j \in \mathcal{N}(x_i)} k(x_i, x_j) f(x_j), \tag{4}$$

where $\mathcal{N}(x_i) \subset \mathcal{P}$ is a neighborhood of $x_i$ and kernel $k$ takes different forms in different methods. PointNet Qi et al. (2017a) is an early attempt to extract features on point cloud. PointNet is a network structure without convolution, or alternatively we can interpret the convolution defined by PointNet has the simplest kernel $k(x_i, x_j) = \delta(x_i, x_j)$ where $\delta$ is the Kronecker-Delta. Various later works attempt to improve PointNet by choosing different forms of the kernel $k$. For example, PointNet++ Qi et al. (2017b) introduces a max pooling among local points, i.e. choosing kernel $k$ as an indicator function: $k(x_i, x_j) = I_{x_j = \arg\max_{z \in \mathcal{N}(x_i)} f(z)}$. Pointcnn Li et al. (2018b) chooses $k(x_i, \cdot) = \beta' A_{x_i}$ where $\beta \in \mathbb{R}^K$ and $A_{x_i} \in \mathbb{R}^{K \times K}$ are trainable variables with $K = |\mathcal{N}(x_i)|$.

The second type of convolution is defined by first projecting the point cloud locally on an Euclidean domain and then employ regular convolution. This type of methods can also be formulate as (2). For example, Tangent convolutions Tatarchenko et al. (2018) define kernels on the tangent plane, and use 2 principal directions of a local PCA as $\vec{u}_x$. Pointconv Wu et al. (2018) constructs local kernels by interpolation in $\mathbb{R}^3$, i.e. letting $\phi(x, v) = x - v$ which is essentially a local Euclidean convolution.

## 1.2 THE PROPOSED CONVOLUTION: NPTC

In this paper, we propose Narrow-Band Parallel Transport Convolution (**NPTC**), which is a geometric convolution based on point cloud discretization of a manifold parallel transport defined in a specific way. As we observed in the previous section, convolutions in many methods can be written in the form of (2) and (3), while the differences mostly lie in the choices of the vector field $\{\vec{u}_x\}_{x \in \mathcal{M}}$. As observed by Schonsheck et al. (2018) that choosing the vector field properly, the associated convolution can be interpreted as parallel transporting the kernels using the parallel transport associated to the prescribed vector field. We propose to define convolution on point clouds by combining narrow-band approach Adalsteinsson & Sethian (1995) and geometric convolution. The formal definition of parallel transport and general ideal will presented in Section 2 and detailed descriptions on NPTC will be given in Section 3.

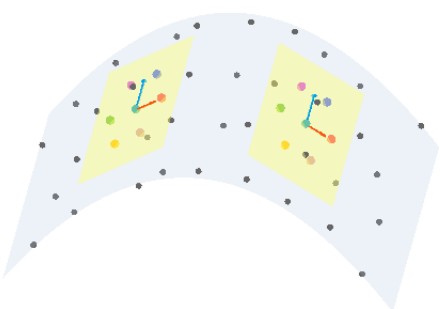

Figure 1: Narrow-band parallel transport convolution on point cloud: black and blue points are sample points of the surface (blue). Each kernel (colored dots) is defined on the tangent plane (yellow). The vectors on the tangent planes that are of the same color are in a parallel fashion.

Now, we describe how NPTC is computed on point clouds. Firstly, point clouds are approximated by voxel narrow-band with appropriate resolution. For a manifold $\mathcal{M}$ or point cloud $\mathcal{P}$, its narrow-band is defined as the region with distance from $\mathcal{M}$ (or $\mathcal{P}$) less than $\epsilon$: $\mathcal{NB}(\mathcal{M}) := \{x \in \mathbb{R}^3 | dist(x, \mathcal{M}) < \epsilon\}$, where $dist$ represents a distance function. For calculation efficiency and robustness to noise, we choose voxel narrow-band approximation. It should be noted that the result of the traditional voxelization method Wu et al. (2015) is a solid 3-d structure and the voxel narrow-band can be understood as the "shell" with a certain thickness of the former (if the point cloud is sampled from the surface of the object). It is very efficient to calculate the distance function in voxel approximation with appropriate resolution. The vector field $\{\vec{u}_x\}_{x \in \mathcal{M}}$ is defined as the projections of the gradient field of a narrow-band-based distance function on the approximated tangent plane of the point cloud. Such definition of the vector field is robust to noise, since if the distortion of the coordinates of the points by noise does not exceed the width of the narrow-band, the

computed gradient field of the distance function in the narrow-band remains unchanged. After the vector field $\{\vec{u}_x\}_{x\in\mathcal{M}}$, we use local PCA to estimate normal vectors on the point cloud following Xu et al. (2018). Finally, the convolution kernel can be constructed on the tangent plane and the corresponding $f(v)$ can be obtained by interpolating the value $f(x)$ on the point cloud.

Note that we prefer to use geometric convolution in NPTC because compared with methods that translate kernels in the ambient space of the manifold, NPTC translates kernels on the tangent planes which effectively avoids having convolution kernels defined away from the underlying manifold of the point cloud. In other words, NPTC can well reflect point cloud geometry and is a natural generalization of planar convolution in the sense that when the point cloud reduces to planar grids, the NPTC reduces to the planar convolution.

### 1.3 Contributions

- We introduce a new point cloud convolution, NPTC, based on parallel transport defined by a narrow-band approximation of the point cloud. The proposed NPTC is a natural generalization of planar convolution.

- The proposed NPTC combines voxelization and geometric convolution. Voxelization with appropriate resolution brings robustness and geometric convolution can better reflect point cloud's geometry.

- Based on NPTC, we designed convolutional neural networks, called NPTC-net, for point clouds classification and segmentation with state-of-the-art performance.

## 2 Background and General Idea

We recommend the book Nakahara (2003) for readers who are unfamiliar with differential geometry. A short intuitive introduction to parallel transport will be provided after recalling the notion.

Let $\mathcal{M}$ be a 2 dimensional differential manifold embedded in $\mathbb{R}^3$ and $\mathcal{T}_x\mathcal{M}$ be the tangent plane at point $x \in \mathcal{M}$. $\mathcal{T}_x\mathcal{M}$ can be defined as the 2-dimensional linear space formed by the span of 2 tangent vectors. The disjoint union of the tangent planes at each point on the manifold defines the tangent bundle $\mathcal{T}\mathcal{M}$. A vector field $\mathbf{X}$ is a section of $\mathcal{T}\mathcal{M}$. Collection of all smooth vector fields $\mathbf{X}(x) \in \mathcal{T}_x\mathcal{M}, \forall x \in \mathcal{M}$ is denoted as $\Gamma(\mathcal{T}\mathcal{M})$. An affine connection is a bilinear mapping $\nabla$: $\Gamma(\mathcal{T}\mathcal{M}) \times \Gamma(\mathcal{T}\mathcal{M}) \to \Gamma(\mathcal{T}\mathcal{M})$. A section of a vector field is called **parallel** along a curve $\gamma$ if $\nabla_{\dot{\gamma}}X = 0$. Suppose we are given an vector $\vec{e} \in \mathcal{T}_{x_0}$ at $x_0 = \gamma(0) \in \mathcal{M}$. The **parallel transport** of $\vec{e}$ along $\gamma$ is the extension of $\vec{e}$ to a parallel section $\mathbf{X}$ on $\gamma$.

Intuitively, in differential geometry parallel transport is a way of translating a vector along a smooth curve "parallel" to the original. In fact, by given smooth vector fields $\{\vec{u}^1, \vec{u}^2\}$, we can define linear transformation among tangent planes then the corresponding parallel transport can be induced Knebelman (1951) Therefore, parallel transport defined in this way can be viewed as parallel transporting the vector on the manifold with respect to the connection that is reconstructed from the vector field $\{\vec{u}_x^1\}_{x\in\mathcal{M}}$. That is to say, if given a vector field, we can define a way to "transport" vectors between tangent planes.

To select a suitable vector field, we first recall the choice of the vector fields of PTC which defines convolution on triangulated surfaces via parallel transport with respect to the Levi-Civita connection Schonsheck et al. (2018). Geodesic curve represents, in some sense, the "straightest" path between two points on a Riemannian manifold. Given a geodesic connecting two points $x$ and $y$, the tangential direction at $x$ corresponds to the ascent direction of geodesic distance from $y$. That is to say, all tangential directions of one geodesic form a parallel vector field. PTC chooses such direction as $\vec{u}_x^1$ and defines $\vec{u}_x^2 = \vec{u}_x^1 \times \vec{n}_x$ with $\vec{n}_x$ the normal vector at $x$. One naive approach to extend mesh based methods to point cloud is to generate a triangulated surface based on the point cloud. However, this is not as convenient as working directly with the point cloud since in practice not every point cloud corresponds to a legitimate parameterized surface and pooling is not as easy to implement on triangulated surfaces as on point clouds. In addition, it is time-consuming to construct meshes on point clouds. When applied in practice, mesh construction time may be much longer than inference time of some methods directly applied on the point cloud.

If we want to construct a vector field on point cloud, gradients of some sort of distance function can be a good choice. However, unlike triangulated surfaces, distance function is not easily defined on point clouds due to the lack of connectivity. It is then natural to approximate the point cloud with another data structure with connectivity, so that distance function can be easily calculated. We use voxel to approximate the point cloud in a narrow-band in $\mathbb{R}^3$ covering the point cloud. We denote such distance function as $\rho : \mathbb{R}^3 \to \mathbb{R}^+$.

Then, we review how a distance function is calculated. Distance functions can be easily computed by solving the Eikonal equation. The Eikonal equation is a non-linear partial differential equation describing wave propagation:

$$|\nabla \rho| = 1/h(x), \quad x \in \Omega, \quad \rho|_\Lambda = 0, \tag{5}$$

where $\Lambda \subset \overline{\Omega} \subset \mathbb{R}^n$ and $h(x)$ is a strictly positive function. The solution $\rho(x)$ of (5) can be viewed as the shortest time needed to travel from $x$ to $\Lambda$ with $h(x)$ being the speed of the wave at $x$. For the special case when $h = 1$, the solution $\rho(x)$ represents the distance from $x$ to $\Lambda$ limited in the $\Omega$. The Eikonal equation can be solved by the fast marching method Sethian (1996).

Note that if the point cloud is sampled from a plane, the narrow-band is flat as well. Then, by a proper choice of the distance function, the vector fields $\{\vec{u}_x^j\}_{x \in \mathcal{M}}, j = 1, 2$, can be reduced to the global coordinate $\{\vec{e}_j\}, j = 1, 2$ on the plane. This means that NPTC is reduced to the traditional planar convolution.

Once the distance function $\rho$ is computed, we choose $\vec{u}_x^1 = \nabla_\mathcal{P} \rho(x)$, where $\nabla_\mathcal{P} \rho(x)$ is a projection of $\nabla \rho(x)$ on an approximated tangent plane at $x$. Then, $\vec{u}_x^2$ can be calculated by the outer product $\vec{u}_x^2 = \vec{u}_x^1 \times \vec{n}_x$ with $\vec{n}_x$ the normal vector at $x$.

For convenience, instead of transporting the kernel on the manifold, we can locally construct the parallel transported kernel at every point $x$ by formulating $k$ as $k(\phi(x, \cdot))$, where $\phi(x, v)$ is defined in (3). It is known in differential geometry that transporting the kernel to every point on the manifold is the same as locally reconstructing the kernel in the aforementioned way.

# 3 Narrow-band Parallel Transport Convolution (NPTC) and Network Design

## 3.1 Implementation of NPTC

Note that some methods of voxelization, interpolation ,and initial point selection are not essential. These can be replaced depending on the accuracy, speed requirements and datasets.

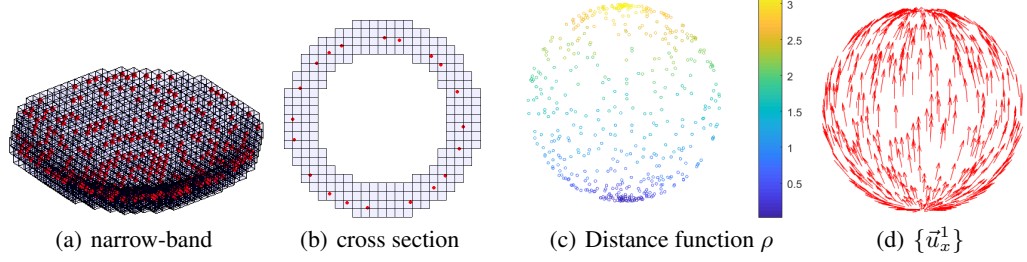

|  (a) narrow-band | (b) cross section | (c) Distance function $\rho$ | (d) $\{\vec{u}_x^1\}$ |

Figure 2: Illustration of a point cloud $\mathcal{P}$ sampled from the unit sphere. (a) shows the narrow-band approximation (blue boxes) of part of $\mathcal{P}$ (in red). (b) is a cross section of (a). (c), (d) show the distance function $\rho$ and vector field $\{\vec{u}_x^1\}$ ($\{\nabla_\mathcal{P} \rho(x)\}$) on the point cloud. We can see that distance propagates from the bottom center to the top center reflecting the geometry of the sphere.

### 3.1.1 Voxelization and Computing Distance Function

Assuming that the point cloud has been normalized so that it is enclosed by the unit cube. The unit cube is then discretized by $N \times N \times N$ grids. The narrow-band approximation of the point cloud is constructed by including each voxel with a distance less than $\frac{k}{N}$ from any point on the point cloud. Then the voxel narrow-band approximation of point cloud has been constructed. $N$ depends on the

requirements of speed and accuracy. $k$ need to be large enough to make the narrow-band continuous (if the point cloud represents a continuous manifold) and small enough to make the distance in narrow-band approximate the geodesic distance. In the next experiments, we choose $N$ as 100, $k$ as 2 and $L^\infty$ as distance function. If the sampling of point cloud is very sparse, it may be necessary to construct a narrow-band by the estimation of the normal or some other strategies.

Then we can compute distance functions on the voxels using the well-known fast marching method based on regular grid provided by the voxelization Sethian (1996). The solution $\rho(x)$ of the Eikonal equation $|\nabla\rho(x)| = 1$ presents the distance form $\Lambda$ to $x$ limited inside the narrow-band. Here $\Lambda$ is chosen as certain point on the point cloud.

### 3.1.2 Computing the Vector Fields on Point Clouds

We first compute the tangent plane on each point. Tangent planes are important features of manifolds and have been well-studied in the literature Lai et al. (2013). In this paper, we use local principal component analysis (LPCA) to estimate the tangent plane. We estimate the local linear structure near $x \in \mathcal{P}$ using the covariance matrix

$$\sum_{x_k \in \mathcal{N}(x)} (x_k - c)^\top (x_k - c), \quad c = \frac{1}{k} \sum_{x_k \in \mathcal{N}(x)} x_k,$$

where $\mathcal{N}(x)$ is the set of neighboring points of $x$. The eigenvectors of the covariance matrix form an orthogonal basis. If the point cloud is sampled from a two dimensional manifold, and the local sampling is dense enough to resolve local features, the eigenvectors corresponding to the largest two eigenvalues provide the two orthogonal directions of the tangent plane, and the remaining vector represents the normal direction at $x \in \mathcal{P}$. Here, we denote the space spanned by the two eigenvectors of the covariance matrix at $x$ as $\mathcal{T}_x\mathcal{P} \subset \mathbb{R}^3$.

With the computed distance function $\rho(x)$, it is natural to define the vector field by projecting $\nabla\rho$ on the approximated tangent planes of the point cloud. Given a point $x_k \in \mathcal{P}$ close enough to $x \in \mathcal{P}$, we have

$$\langle \nabla\rho, x_k - x \rangle \approx \rho(x_k) - \rho(x),$$

where $\rho(x_k)$ and $\rho(x)$ are known. If we consider $k$-nearest neighbors of $x$, we have $k-1$ equations with 3 unknowns that are the three components of $\nabla\rho$. We can use least squares to find $\nabla\rho$. We then project the vector $\nabla\rho(x)$ onto the tangent plane at $x$. We denote the projected vector $\nabla_{\mathcal{P}}\rho$, which is the vector we eventually need to define NPTC.

After defining the vector field on the tangent plane, we choose the convolution kernel of hexagon (Figure 1), that is, a convolution kernel is composed of a central point and six evenly distributed points around. The value $f(v)$ is computed by $f(v) = f(z)$ where $z \in \mathcal{P}$ is the closest point to $v$ in $\mathbb{R}^3$. Note that one may use a more sophisticated method to compute $f(v)$ rather than using the closest point interpolation. We choose the closest point interpolation because of its simplicity.

### 3.2 NPTC-net: Architecture Design for Classification and Segmentation

This section, we present how to use NPTC to design convolutional neural networks on point clouds for classification and segmentation tasks. For that, other than the NPTC, we need to define some other operations that are frequently used in neural networks. Note that, some point-wise operations like MLP and ReLu are the same on point cloud as the Euclidean case. Here, we only focus on the operations that are not readily defined on point clouds.

**Down-sampling:** In our implementation, the sub-sampled set of points of the next layer is generated by the farthest point sampling Eldar et al. (1997).

**Convolution layer:** Our $k$-th convolution layer takes points $\mathcal{P}^k \in \mathbb{R}^{N^k \times 3}$ and their corresponding feature maps $\mathcal{F}^k \in \mathbb{R}^{N^k \times c^k}$ as input, where $N^k$ is the number of the points and $c^k$ is the number of channels at layer $k$. The corresponding output is $\mathcal{F}^{k+1} \in \mathbb{R}^{N^{k+1} \times c^{k+1}}$ living on the points $\mathcal{P}^{k+1} \in \mathbb{R}^{N^{k+1} \times 3}$. The NPTC-net have encoding and decoding stages. Normally, $N^{k+1} < N^k$ during encoding and $N^k < N^{k+1}$ during decoding. Convolution at the $k$-th layer is only performed on the point set $\mathcal{P}^{k+1}$, which resembles convolution with stride $> 1$ for planar convolutions.

**Residual block:** One residual block takes the feature maps $\mathcal{F} \in \mathbb{R}^{N \times c}$ on the point set $\mathcal{P} \in \mathbb{R}^{N \times 3}$ as input and same number of points and same number of channels of features as output. One residual layer consists of three components: MLP from $c$ channels to $\frac{c}{2}$ channels, convolution layer from $\frac{c}{2}$ channels to $\frac{c}{2}$ channels, MLP from $\frac{c}{2}$ channels to $c$ channels plus the feature maps from the bypass connection. A residual block consists of several residual layers.

**NPTC-net** consists of the aforementioned operations and its architecture is given by Figure 3. The left half of the NPTC-net is the encoder part of the network for feature extraction. For classification, features at the bottom of the network are directly attached to a classification network; while for segmentation, features are decoded using the right half of the NPTC-net (decoder part of the network) to output the segmentation map.

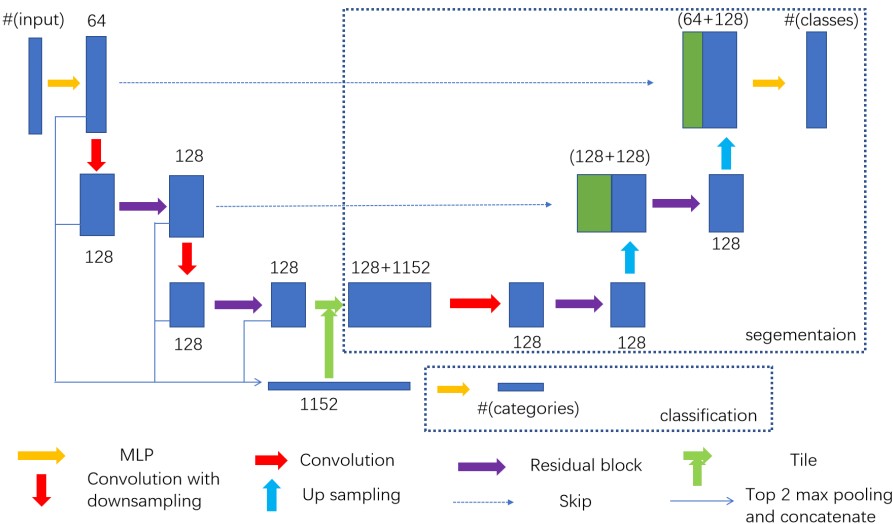

Figure 3: Architecture of NPTC-net. The network for segmentation tasks is at the top right with encoding and the network for classification tasks is at the bottom right with encoding. Top 2 max pooling means taking the maximum two values among all the points in each channel and then concatenate into one vector representing global feature. "Tile" means repeating the global feature and concatenating to each point as extra channels.

## 4 EXPERIMENTS

In order to evaluate our new NPTC-nets, we conduct experiments on three widely used 3D datasets, ModelNet Wu et al. (2015), ShapeNet Yi et al. (2016), S3DIS Armeni et al. (2016). We implement the model on a GTX TITAN Xp GPU. For more implementation details, please refer to the Appendix.

### 4.1 3D SHAPE CLASSIFICATION AND SEGMENTATION

We test the NPTC-net on ModelNet40 for classification tasks. ModelNet40 contains 12,311 CAD models from 40 categories with 9,842 samples for training and 2,468 samples for testing. For comparison, we use the data provided by Qi et al. (2017a) sampling 2,048 points uniformly and computing the normal vectors from the mesh. As shown on Tabel 1, our networks outperform other state-of-art methods. (If a compared method has results on both 2048 (or 1024) and 5000 points, we only compare with the former.).

We also evaluate the NPTC-net on ShapeNet Part for segmentation tasks. ShapeNet Part contains 16,680 models from 16 shape categories with 14,006 for training and 2,874 for testing, each annotated with 2 to 6 parts and there are 50 different parts in total. We follow the experiment setup of previous works, putting object category into networks as known information. We use point intersection-over-union (IoU) to evaluate our NPTC-net. Table 1 shows that our model ranks second on this dataset and is fairly close to the best known result.

Table 1: Comparisons of overall accuracy (OA) and mean per-class accuracy (mA) on ModelNet40 as well as comparisons in instance average IoU (mIoU) and class average IoU (mcIoU) on ShapeNet Part. Models ranking first is colored in red and second in blue.

| | Modelnet40 | | ShapeNet part | |
| --- | --- | --- | --- | --- |
| Method | OA(%) | mA(%) | mIoU | mcIoU |
| kd-net Klokov & Lempitsky (2017) | 91.8 | 88.5 | 82.3 | 77.4 |
| pointnet Qi et al. (2017a) | 89.2 | 86.2 | 83.7 | 80.4 |
| SO-Net Li et al. (2018a) | 90.9 | 87.3 | 84.9 | 81.0 |
| pointnet++ Qi et al. (2017b) | 90.7 | - | 85.1 | 81.9 |
| SpecGCN Wang et al. (2018a) | 92.1 | - | 85.4 | - |
| SpiderCNN Xu et al. (2018) | 92.4 | - | 85.3 | 81.7 |
| pointcnn Li et al. (2018b) | 92.2 | 88.1 | 86.1 | 84.6 |
| DGCNN Wang et al. (2018c) | 92.2 | 90.2 | 85.1 | 82.3 |
| Ours | 92.7 | 90.2 | 85.8 | 83.3 |

## 4.2 RUBOSTNESS TEST

As pointed out earlier, using voxelization can bring rubustness to the definition of geometric convolution. In order to show the robustness of our model for real data, we tested scene semantic segmentation on the "Stanford Large-Scale 3D Indoor Spaces Dataset" (S3DIS). S3DIS covers six large-scale indoor areas from 3 different buildings for a total of 273 million points annotated with 13 classes. This is a real-word scanned dataset without normal and with noise. Following Tchapmi et al. (2017), we advocate the use of Area-5 as test scene to better measure the generalization ability of our method. Table 2 shows that geometric convolution methods are close to the methods which do not need normal estimation. NPTC-net outperforms the best known geometric convolution method TangentConvXu et al. (2018).

Table 2: Comparisons of overall accuracy (OA) and mean per-class IoU (mIoU) on S3DIS. Models ranking first is colored in red and second in blue.

| Convolution Type | Method | OA(%) | mIoU(%) |
| --- | --- | --- | --- |
| no convolution | pointnet Qi et al. (2017a) | 78.8 | 41.3 |
| 3-d convolution | SegCloud Tchapmi et al. (2017) | - | 48.9 |
| | Eff3DConvZhang et al. (2018) | 69.3 | 51.8 |
| | ParamConvWang et al. (2018b) | - | 58.3 |
| geometric convolution | TangentConv Xu et al. (2018) | 82.5 | 52.8 |
| | Ours | 83.7 | 54.0 |

## 5 CONCLUSION

This paper proposed a new way of defining convolution on point clouds, called the narrow-band parallel transport convolution (NPTC), based on a point cloud discretization of a manifold parallel transport. The parallel transport was defined specifically by a vector field generated by the gradient field of a distance function on a narrow-band approximation of the point cloud. The NPTC was used to design a convolutional neural network (NPTC-net) for point cloud classification and segmentation. Comparisons with state-of-the-art methods indicated that the proposed NPTC-net is competitive with the best existing methods.

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

## A APPENDIX

### A.1 IMPLEMENTATION DETAILS

We implement the model with Tensorflow using SGD optimizer with an initial learning rate 0.1 for ModelNet40 and ADAM optimizer with an initial learning rate 0.002 for ShapeNet Part and S3DIS on a GTX TITAN Xp GPU. During the training procedure of classification, the data is augmented by random rotation, scaling and Gaussian perturbation on the coordinates of the points. During the training procedure of segmentation, the data is augmented by scaling and Gaussian perturbation on the coordinates of the points.

Following Li et al. (2018b), the data of S3DIS is firstly split by room, and then the rooms are sliced into 1.5m by 1.5m blocks, with 0.3m padding on each side. The points in the padding areas serve as context of the internal points, and themselves are not linked to loss in the training phase, nor used for prediction in the testing phase. Each block will be viewed as a point cloud during training and testing.

We do inferences on the augmented data during testing by voting following Qi et al. (2017b). To clarify this, let $x^i$ be one data from the test set and $x^i_j$ be the augmented data generated from $x^i$. $p^i_j = f(x^i_j)$ denotes the predict possibility vector after Softmax. The final inference result of $x^i$ is $\text{Predict}(x^i) = \arg\max_k (\sum_j p^i_j)_k$.

### A.2 RUNNING STATISTICS

Table 3: Comparisons of number of parameters and FLOPs for classification.

| method | Parameters | FLOPs(Inference) |
|---|---|---|
| pointnet Qi et al. (2017a) | 3.48M | 14.70B |
| pointnet++ Qi et al. (2017b) | 1.48M | 26.94B |
| DGCNN Wang et al. (2018c) | 1.84M | 44.27B |
| pointcnn Li et al. (2018b) | **0.6M** | 25.30B |
| Ours | 1.29M | **11.7B** |

Table 4: Comparisons of training time of networks on ModelNet40

| Method | Settings | Accuracy and Training (+Pre-processing) time |
|---|---|---|
| PointNet++ Qi et al. (2017b) | adam, 1024 points | 90.7%, 6 hours |
| | adam, 5000 points | 91.9%, 20 hours |
| ours | adam, 2048 points | 92.4%, 6h (+1.5h) |
| | SGD, 2048 points | 92.7%, 12h (+1.5h) |

As shown in Table 3, we summarize our running statistics based with model for ModelNet40 with batch size 16. In comparison with several other methods, although we use ResNet structure, the fewer channels, smaller kernels and simpler interpolation (nearest neighboring) make NPTC use similiar parameters and even fewer FLOPs.

Total pre-processing time mainly depends on the grid's density. For most cases, the resolution of $100^3$ is enough to describe the shape of the point cloud, which is what we chose for our experiments.

We also remark that the whole computation cost of constructing convolution is linearly depending on the resolution of voxel and the size of data. On a PC with Core i7-7700 CPU, the pre-processing takes about **0.5** seconds per point cloud with Matlab, and the whole dataset of ModelNet40 (12,311 shapes with 10,000 points each) takes only about 1.5 hours. It is negligible compared to the training of the deep neural networks and acceptable to do inference in practice.

## A.3 VISUALIZATION

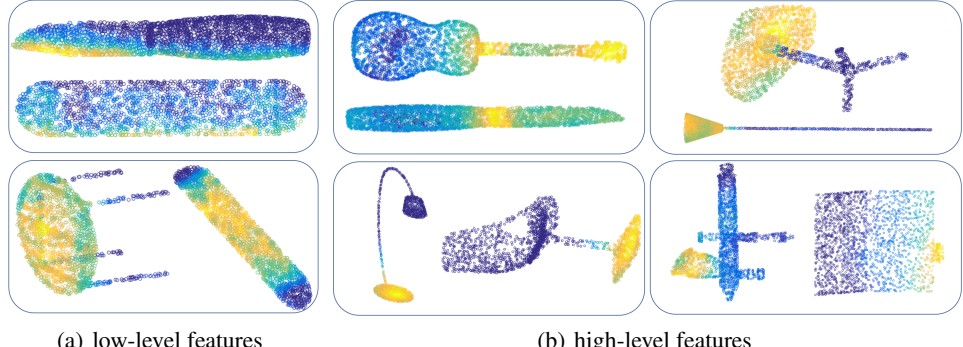

(a) low-level features                    (b) high-level features

Figure 4: Feature Visualization: each feature from low or high level is displayed on 2 point clouds from different categories. High-level activations are in yellow and low-level activations are in blue.

To visualize the effects of the proposed NPTC in the NPTC-net, we trained the network on ShapeNet Part and visualize learned features by coloring the points according to their level of activation. In Figure 4, filters from the the first Convolution layer in the the first Residual block and final Convolution layer in the second Residual block are chosen. In order to easily compare the features at different levels, we interpolate them on the input point cloud. Observe that low-level features mostly represent simple structures like edges (top of (a)) and planes (bottom of (a)) with low variation in their magnitudes. In deeper layers, features are richer and more distinct from each other, like bottleneck (upper left of (b)), "big-head"(upper right of (b)), plane base (lower left of (b)), bulge (lower right of (b)).

## A.4 SINGULARITIES

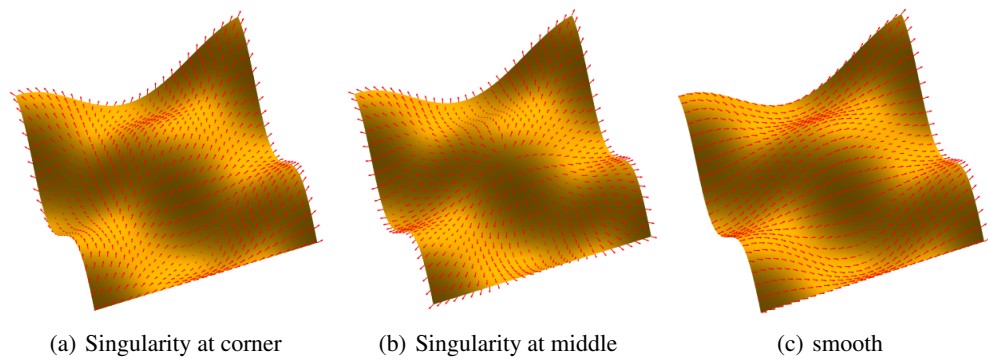

(a) Singularity at corner        (b) Singularity at middle              (c) smooth

Figure 5: Different strategies induce different vector fields: (a) is the vector field which selects the corner as the origin of the distance function. (b) is the vector field which selects the middle as the origin. (c) is the linear combination of two vector fields with different origins. The combination coefficients depend on the distance to the origin at each point: $\vec{v}_x = d(y_1, x)\vec{v}_x^1 - d(y_2, x)\vec{v}_x^2$. $\{\vec{v}^i\}$ represent the vector field with the origin $y_i$ and $d(y_i, x)$ represent the geodesic distance. The absolute value of combination coefficient for one vector field will descend to $0$ as $x$ approaches the origin to prevent the singularity at the origin.

We note that on non-parallelizable manifolds such as the sphere, it is not possible to construct a smooth vector field. Helgason (1979) So that the vector field must have at least one a singularity,

and the frame around the singularity will rotate violently. Figure 5 shows vector fields defined on the surface with and without singularity.

If singularity can not be avoided. It is natural to ask how much singularities affect the result. This has been discussed in section 4.4 of PTC. Schonsheck et al. (2018) In surface MNIST experiment, convolutions defined by taking the middle point of a surface as the origin of the distance function (which means that there is a singularity in the middle like Figure 5 (b)) will reach the accuracy of 94.92% and convolutions using the smooth strategy will reach 96.36%. Singularity does cause some problems, but we think it's in the acceptable range. In our experience, activation in segmentation task have been visualized as before, we find no direct relationship with the location of the singularity.

If we have some prior knowledge about a shape that can have no singularity. Strategies can be applied to get a smooth vector field or reduce the influence of singularity. For example, if the shape is essentially like a wrinkled sheet, then we can choose the starting point(s) $\Lambda$ of the Eikonal equation as points on one edge of the sheet or the linear combination of two vector fields with different origins Fig 5 (c).

Most of the shapes in the three experiment datasets are closed manifolds like boxes, beds, airplanes, on which singularity can not be avoided. Experiments show that applying more strategies we know can not improve the results obviously, so we choose to select one initial point directly.

