# OpenReview forum: "NPTC-net: Narrow-Band Parallel Transport Convolutional Neural Network on Point Clouds"
_ICLR.cc/2020/Conference — Reject_

### Official Review · AnonReviewer1 · 2019-10-16
**Official Blind Review #1**

**Rating:** 3

**Review:**

The paper introduces a new geometric convolution that can be applied to point clouds approximating a 2D manifold in R^3. Similar to previous work, the continuous mathematical description of the method involves a kernel that is a function on R^2, which is pushed to the tangent spaces of the manifold via a choice of frame u. A local neighbourhood around the kernel is parameterized by the tangent space via the exponential map and matched against the kernel. The frame u, which determines the orientation of the kernel, is chosen by first picking an origin and then setting u1 to the direction of steepest ascent of the function that measures geodesic distance to the origin. The other basis vector u2 is computed as u1 x n where n is the normal vector. In a practical implementation, the point cloud is voxelized in order to compute the distance function and it's gradient. The gradient is then interpolated back to the point cloud. If I understood correctly, a convolution is then directly applied to the point cloud, though I could not find a description of how this is done.

The problem of defining geometric convolutions on point clouds is interesting and relevant, and something like what is presented in this paper might work well. Indeed the experiments demonstrate that the method produces decent scores on modelnet and S3DIS. The method is not very fast, due to the voxelization preprocessing which is required, which at .5s precludes real-time applications. Also, as discussed below, there are some issues related to singularities in the vector fields u1,u2 that are not adequately discussed. My main concern, and the reason I have given a "weak reject" rating, is that the clarity and completeness of the paper is insufficient and I find the paper hard to follow (see below). I would actually like to see an improved version published, and if the authors can address the concerns I would increase my score.


# Writing
I find that the paper is hard to read because of the way it is structured, as well as numerous small mathematical and expository issues (discussed below) and missing information.

The present paper builds on the PTC method of Schonsheck et al., but this method is not properly explained in the paper. PTC is mentioned in 1.1, but only described at a very high level. Since I had not read this paper before (I have now) the present paper was hard to follow on a first reading.

Section 1.2 is a large sub-section of the related work section that actually discusses the proposed NPTC. I think it's better not to spend so much of the related work section on this. Moreover, section 3.1.1. "General idea of NPTC", contains another high level description, which is somewhat redundant (though at the same time, both sections leave lots of questions unanswered - see below). The following sections, 3.1.2 and 3.1.3 explain the computation of the distance function and the vector fields, but contain no clear explanation of the whole algorithm.

Many important questions about how the method is actually implemented are left unanswered, even though the main contribution of this paper is at the implementation level rather than the theoretical level.
- How is voxelization done? Count the number of points per voxel? How is the voxel size chosen? How do you make sure that you obtain a "narrow band" around the surface defined by the point cloud? What kind of datastructure do you use to store a sparse voxel grid?
- How is the filter parameterized? In the theory, k is a function on R^2. How do we evaluate it at an arbitrary point in R^2?
- How is convolution done? I think it is done on points not voxels, similar to eq. 4, but this is never stated in the paper. How do you approximate the integral over the tangent space in Eq 2, if eq 2 is what the conv is based on?

I would remove section 2.1 where the notion of a connection is formally introduced. The reader will either know this concept, in which case it is unnecessary, or not, in which case the description is far too compact (one must read a textbook to really understand this notion). It may be better to include only a short intuitive introduction to parallel transport, refering the reader to other texts for mathematical definitions of basic notions like connections.

The fact that the voxelization of the point cloud is sparse is only mentioned very late in the paper. For a while I was thinking the idea would be to construct a dense voxel grid, which would be extremely memory intensive. Also, the phrase "narrow-band" is often used with a very different meaning, as in a small part of the spectrum. It was not clear to me what this was referring to until halfway through the paper. Better to explain the general idea of the algorithm in the introduction.


# Singularities
I looked at the papers citing the main related work (the PTC paper by Schonscheck et al., 2018), and of the four listed in google scholar, one seems relevant in that it prominently cites Schonscheck et al. and claims to improve upon it, but this work is not cited in the present paper:
Cohen, Weiler, Kicanaoglu, Welling, Gauge Equivalent Convolutional Networks and the Icosahedral CNN.
This work highlights one potential downside of the PTC approach, which is that it is not possible to find a global frame u (called in w in their paper) without singularities on manifolds like the sphere. Moreover, the choice of frame is fundamentally arbitrary and should not affect the results. In the PTC approach, a certain choice of frame with singularities is made, which leads to filter orientations that are 1) essentially arbitrary, and 2) change smoothly in most places but abruptly near the singularities. The results are heavily dependent on the choice of frame.

The present paper dismisses these issues by saying that "We remark that possible singularities will lead to no convolution operation at those points. These are isolated points on a closed manifold and do not effect experiment results."
However, this is not actually demonstrated in the paper, and one might reasonably believe that this problem affects more than a small finite set of points. Near a singularity (e.g. the poles of the sphere in fig 2d), nearby filters will be oriented very differently. If we draw a small circle around the singularity, the orientation of the filter changes 360 degrees as we travel around the circle. As a result, the meaning of the patterns in the output feature map is very different near the poles than it is near the equator. Nevertheless, the next layer will try to match these patterns with the same set of filters / perform weight sharing across different positions. In my opinion this issue should be more frankly discussed.

Fortunately, in some applications with point clouds this issue can be avoided. For point clouds coming directly from a (RGB+)depth sensor, the manifold is essentially like a wrinkeled sheet (homeomorphic to a disk), and one can easily find vector fields / frames without singularities on such manifolds. (this is not necessarily true for point clouds that represent a whole scene, e.g. in SLAM). I think this is what is mentioned in paragraph 3 of 3.1.1. However, it is not clear to me if the frame used in this paper is without singularities even when it is possible to find such a frame. The reason is that the frame is defined in terms of the distance function rho to an arbitrary origin. If I understand correctly, one will always have a singularity at the origin. This could in principle easily be solved though by simply using the x and y direction of the camera plane as a frame (projecting it from R^3 to each tangent space). However, if the authors decide to go that route, one would also like to see a comparison to simply applying 2d convolutions to the RGB+D images.


# Experiments

The experiments on Modelnet40 show that, when compared with other point-cloud methods, the proposed method outperforms. I do not know if there are better point-based methods than those listed in the paper, but on the official Modelnet40 leaderboard (which is incomplete) there are several methods that score better (97%+). Since this difference could possibly be attributed to a loss of information due to the discretization into a point cloud, I still think these results are quite good. Similarly on S3DIS the method scores quite well, though does not reach state of the art.

I could not find the architecture for the classification network, and only a high-level description of the segmentation architecture. More details on the training and evaluation procedures could be added for both experiments.


# Misc

Some more things I noticed below. Many of these are not serious problems, just things that can be improved / fixed.

- p2:
-- "Spatial mesh-based methods are more intuitive" - this is subjective
-- f(v)  where v is a tangent vector in T_xe M is not defined. One would expect f to be a function on the manifold. Typically one writes f(exp_x v) instead.
-- script M is not defined (though it is intuitive that this is the manifold). Later in 2.1 it is written "Let M be a 2 dimensional differential manifold". Later still (3.1) the notation is switched to P without notice.
-- TxM is not defined. Later it is explained this is the tangent plane. Neither is g_x (it is obvious this should be the metric, but only if one knows differential geometry).
-- "and do not effect experimental results" -> affect
-- "singularities from a given vector field can be overcome ...." I don't understand what this is saying. How can we avoid the issues with singularities?
-- consider writing log instead of exp^{-1}
-- "while uses tangent planes" -> using
- p3:
-- You have f(z) in the subscript of the indicator I_argmax ...
-- In the explanation of pointnet it is not clear where there are any parameters in the kernel k if k(xi, xj) = delta(xi, xj)
-- I don't understand the explanation of edge convolution. If f(x_j) = 1 (for all j(?)) then k(xi, xj) = MLP(f(xi), f(xi) - f(xj)) would equal MLP(1, 0) which makes no sense.
-- Description of figure 1: "are in parallel": need to say along what curve they are parallel. If I understand correctly, they are only in parallel if they are attached to different points on a minimizing geodesic emanating from the origin.
- p4:
-- "embed in R^3" -> embedded
-- In 2.1 it is explained that T_x M is the tangent plane, but you've already discussed T_x M several times before.
-- A vector field is not just any assignment M -> TM. It is a section of TM, ie a map M -> TM that, when composed with the projection TM -> M, yields the identity on M.
-- The interval "I" mentioned below eq 5 has not been defined. "I" was used as indicator function before.
-- "X is the unique section of Gamma(TM)". The concept of "section" has not been introduced. Moreover, one could say "section in Gamma(TM)" or "section of TM" but not "section of Gamma(TM)" since Gamma(TM) *is* the set of sections of TM. Probably best to say "unique vector field".
-- "there will be a geodesic connecting x0 and x1" - only true if M is connected.
- p5:
-- "how a distance functions are" -> function
-- "easliy"
-- below eq 6 "f(x) is a strictly positive function", but f(x) does not appear in eq 6
-- in 3.1., script P has not been defined. I first thought it is just a new letter for M, but I think it is the point cloud? Needs to be made explicit.
-- "Geodesic curve represents, in some snse, the shortest path". This is false. A geodesic is a *straightest* curve, but a geodesic need not be distance minimizing in any sense.
-- "ascend direction" -> ascent
-- In the first paragraph of 3.1.1 it is not mentioned explicitly that y plays the role of an "origin". This makes the last sentence confusing.
-- "The value f(v) is computed by f(v) = f(z) where z in P is the closest point to v". Firstly, this sentence is in the middle of a paragraph about the vector fields u1,u2, but seems unrelated. Secondly, what are v and z? I guess v is a point in M and z in P? Or is v a tangent vector? In that case, do you mean closest in the embedding space R^3?
- p6:
-- "consider point clouds in R^3". This assumption was already made before, right?
-- "it is not straightforwardly compute distance function" -> "straightforward to compute the distance function"
-- "Here Lambda is chosen as certain point on the point cloud" -> "a certain point". Also, Lambda was used before as a subset of Omega-bar (below eq 6), but here it is a point. I guess this point is the origin. In other places (earlier) you also refer to the origin, e.g. by y. Needs to be made consistent.
-- "Show that the directly" - remove "the"
-- "Finally, we interpolate the distance function from the voxels to point cloud" -> "the point cloud". On first reading, this (page 6!) was the point where I realized (guessed) that you are going to define the convolution on the point cloud not voxel grid. This needs to be explained much earlier.
-- "it is nature to" -> natural
- p7
-- "in our implement" -> implementation
-- "Appendix." -> "Appendix A" or "the Appendix."
- p8
-- "Scene semantics segmentation" -> semantic


**Experience Assessment:**

I have published in this field for several years.

**Review Assessment: Checking Correctness Of Derivations And Theory:**

I carefully checked the derivations and theory.

**Review Assessment: Checking Correctness Of Experiments:**

I carefully checked the experiments.

**Review Assessment: Thoroughness In Paper Reading:**

I read the paper thoroughly.

---

> ### Author Response · Authors · 2019-11-15
> **Response to Review #1**
>
> Review 1：
> We are grateful for the extraordinary detailed and careful comments. Here is our response:
>
> # Writing
> We clarify the writing by elaborating more intuitive descriptions of PTC and our mode. Also, more precise and detailed descriptions of some easily misunderstood concepts are provided at the beginning, such as “voxelization” and “narrow-band”, to prevent the article from becoming memory intensive.  (If M is a k-d manifold embedding in R^n, narrowband of M is a set in R^n: NB(M)={x in R^n|dist(x,M)<e}, where dist() represents some distance function. Voxelization in our method is a voxel approximation of NB(M).) Details of voxelization and convolution have been added to the paper.
>
> Note that some methods of voxelization, interpolation, and initial point selection are not essential. These can be replaced depending on the accuracy, speed requirements and datasets.
>
>
> # Singularities
> Singularities are indeed a problem worthy of discussion. We tend to discuss it in two aspects: if singularities can not be avoided, is this method with singularities meaningful and practical? If singularities can be avoided, can this method avoid? (This part has been added in the appendix of the article, and more experiments will be added in the future version to verify our statement.)
>
> First, most of the shapes in the three experiment datasets are closed manifolds like boxes, beds, airplanes where it is not possible to find a global frame u without singularities. So, the main question is how much singularities affect the result. This has been discussed in section 4.4 of PTC. In surface MNIST experiment, convolutions defined by taking the middle point of a surface as the origin (which means that there is a singularity in the middle) will reach the accuracy of 94.92% and convolutions using the smooth strategy will reach 96.36%. Singularity does cause some problems, but we think it's in the acceptable range. In our experience, activation (Appendix Visualization) and error parts (Not shown) in segmentation task have been visualized, we find that there is no direct relationship with the location of the singularity. More exploration along this direction will be investigated in our future work.
>
> Secondly, dealing with some specific manifold which can avoid singularities, for example, essentially like a wrinkled sheet, then we can choose the starting point(s) (Lambda) (which can be a point sets) of the Eikonal equation as points on one edge of the sheet. (Demos and more strategies can be found in the Appendix.) This is why we say that the selection of the vector field is flexible and some of our implementation details are only for the current datasets. It shows that more strategies we know can not improve the results obviously, so we choose to select one initial point directly.
>
> Moreover, the method of projecting the coordinate system (or some smooth vector field) of the original Euclidean space to the tangent plane as u has been tried before. The result of the classification task is similar to that of NPTC, but there is a certain gap between NPTC in scene semantic recognition. We think that in the complex structure, u defined by intrinsic features will have better results.
>
>
> #Experience:
> We only compared with other point-cloud methods. Taking ModelNet40 as an example, some of the best results are based on “multi-view”. The basic idea of these methods is to project the shape from multiple angles and from multiple 2-D images. The results are obtained by the ensemble method doing classification on the images. These methods are good for extracting global information. However, because of occlusion and other reasons, it can not be directly used in segmentation tasks. Point-cloud methods can be directly used in segmentation tasks.
>
> The architectures for the classification and segmentation are together in Figure 3. This is to show that our model can be easily transplanted to various tasks like traditional CNNs:  “encode + FC” is for classification and “encode + decode” is for segmentation. We updated the description in Figure 3 to make it easier to understand.
>
> # Misc
> All the Misc has been fixed.

---

### Official Review · AnonReviewer2 · 2019-10-23
**Official Blind Review #2**

**Rating:** 3

**Review:**

The authors propose NPTC a new convolutional operator for 3D point clouds embedded on a 2D manifold based no parallel transport defined by a narrow-band approximation. The method combines voxelization within a local neighborhood in 3D (narrow band) and geometric convolution. Experimental results are presneted for a range of classification and segmentation tasks that are similar to the state-of-the-art.
As I am not familiar with geometric methods, point clouds and related work, I wouldn't be able to judge the degree of novelty, beyond an educated guess.

The experimental results in classification and segmentation use the proposed NPTC in standard NN architectures with residual blocks similar to a U-net. In terms of performance, NPTC are amongst the best reported results for each experiment, but always similar to other architectures. Only mean values for IoU and accuracies are reported, but no estimates of variance/spread. Therefore, I cannot judge if the results are significantly different from related methods.

The paper starts out highly technical.
The Notation is introduced in section 2, the Background section, which is after the discussion of related work in section 1, where the notation is already used without being introduced.
The related work makes heavy use of manifold geometry. For non-experts in geometry like me, who also doesn't know about notational conventions, it was near impossible to follow on a first read of the paper. I recommend introduction of notation before use, especially as Section 1 is titled "Introduction".


**Experience Assessment:**

I do not know much about this area.

**Review Assessment: Checking Correctness Of Derivations And Theory:**

I did not assess the derivations or theory.

**Review Assessment: Checking Correctness Of Experiments:**

I assessed the sensibility of the experiments.

**Review Assessment: Thoroughness In Paper Reading:**

I made a quick assessment of this paper.

---

> ### Author Response · Authors · 2019-11-15
> **Response to Review #2**
>
> Thank you for reading and suggestions.
>
> We modified Section 1 and defined the notations before use. To make it easier to understand, we provide a more intuitive point of view in Section 2 to explain what we are doing.
>
> Our results may be similar to the SOTA, but we hope to encourage more explorations about the geometric methods, which are not common at present, especially in point cloud tasks. It may be because the various geometric features on the point cloud are not as easy to calculate as on mesh.

---

### Official Review · AnonReviewer3 · 2019-10-24
**Official Blind Review #3**

**Rating:** 3

**Review:**

This paper proposes Narrow-Band Parallel Transport Convolution (NPTC) for point cloud data. The general idea is to use gradients of some distance function to define the vector field. The authors use voxelization to approximate the point cloud in a narrow-band covering the point cloud so that distance function can be calculated. The authors also discuss how to compute the distance function and the vector fields on point clouds. Experimental results on classification tasks and segmentation tasks are provided.


I am not an expert on this topic and do not have enough expertise to judge the theoretical novelty of this paper (and thus my low confidence). However, I believe the writing can be significantly improved by making the introduction part more friendly to readers unfamiliar with differential geometry.

**Experience Assessment:**

I do not know much about this area.

**Review Assessment: Checking Correctness Of Derivations And Theory:**

I did not assess the derivations or theory.

**Review Assessment: Checking Correctness Of Experiments:**

I assessed the sensibility of the experiments.

**Review Assessment: Thoroughness In Paper Reading:**

I made a quick assessment of this paper.

---

> ### Author Response · Authors · 2019-11-15
> **Response to Review #3**
>
> Thank you for reading and suggestions.
>
> We have revised the structure of our paper. More precise and detailed descriptions of some easily misunderstood concepts are provided. We clarify the writing by elaborating more intuitive descriptions of PTC and our mode in Section 2.

---

### Decision · Program_Chairs · 2019-12-19

**Decision:**

Reject

**Comment:**

All the reviewers recommend rejecting the paper. There is no basis for acceptance.